# Injectable Hydrogel Membrane for Guided Bone Regeneration

**DOI:** 10.3390/bioengineering10010094

**Published:** 2023-01-10

**Authors:** Pauline Marie Chichiricco, Pietro Matricardi, Bruno Colaço, Pedro Gomes, Christine Jérôme, Julie Lesoeur, Joëlle Veziers, Gildas Réthoré, Pierre Weiss, Xavier Struillou, Catherine Le Visage

**Affiliations:** 1CESAM Research Unit, Center for Education and Research on Macromolecules (CERM), University of Liège, B-4000 Liège, Belgium; 2Nantes Université, Oniris, Univ Angers, INSERM, Regenerative Medicine and Skeleton, RMeS, UMR 1229, F-44000 Nantes, France; 3Department of Drug Chemistry and Technologies, Sapienza University, 000185 Rome, Italy; 4Department of Animal Sciences, Animal and Veterinary Research Centre, University of Trás-os-Montes and Alto Douro, 500-801 Vila Real, Portugal; 5FMDUP, Laboratory for Bone Metabolism and Regeneration, Faculty of Dental Medicine, University of Porto, 500-801 Porto, Portugal

**Keywords:** photo-crosslinking, visible light photopolymerization, silanized hydroxypropyl methylcellulose, dextran methacrylate, riboflavin, calvaria bone regeneration

## Abstract

In recent years, multicomponent hydrogels such as interpenetrating polymer networks (IPNs) have emerged as innovative biomaterials due to the synergistic combination of the properties of each network. We hypothesized that an innovative non-animal IPN hydrogel combining self-setting silanized hydroxypropyl methylcellulose (Si-HPMC) with photochemically cross-linkable dextran methacrylate (DexMA) could be a valid alternative to porcine collagen membranes in guided bone regeneration. Calvaria critical-size defects in rabbits were filled with synthetic biphasic calcium phosphate granules in conjunction with Si-HPMC; DexMA; or Si-HPMC/DexMA experimental membranes; and in a control group with a porcine collagen membrane. The synergistic effect obtained by interpenetration of the two polymer networks improved the physicochemical properties, and the gel point under visible light was reached instantaneously. Neutral red staining of murine L929 fibroblasts confirmed the cytocompatibility of the IPN. At 8 weeks, the photo-crosslinked membranes induced a similar degree of mineral deposition in the calvaria defects compared to the positive control, with 30.5 ± 5.2% for the IPN and 34.3 ± 8.2% for the collagen membrane. The barrier effect appeared to be similar in the IPN test group compared with the collagen membrane. In conclusion, this novel, easy-to-handle and apply, photochemically cross-linkable IPN hydrogel is an excellent non-animal alternative to porcine collagen membrane in guided bone regeneration procedures.

## 1. Introduction

Injectable hydrogels that can form in situ have been investigated, with the aim of developing minimally invasive surgical procedures. These 3D networks, composed of biocompatible polymers and reagents, are usually formed under mild conditions (e.g., body temperature, aqueous environment) [1,2]. In recent years, multicomponent hydrogels such as interpenetrating polymer networks (IPNs), which are systems composed of two or more networks, have emerged as innovative biomaterials. The success of the IPN strategy is due to the synergistic combination of favorable properties of each of the polymer networks [3,4,5]. Polysaccharides represent a particularly interesting class of macromolecules for IPN design as they are abundant and available from renewable sources. In addition, they exhibit a large variety of compositions and properties, and they are amenable to tailored chemical modifications to customize the biological/mechanical properties of the resulting hydrogels [6,7,8].

Guided bone regeneration (GBR) is an established clinical procedure in which a bone graft biomaterial is implanted into the wound cavity to trigger bone regeneration, and a membrane is then placed over the bone graft to prevent soft tissue invasion and to enhance the stability of the bone graft material [9,10,11]. The membrane plays a key role in preventing unwanted soft tissue migration into the defect area, and it, therefore, provides sufficient time and space for invasion and proliferation of osteogenic cells.

There is currently a wide range of commercially available membranes. Among these, high-density polytetrafluoroethylene (d-PTFE) membranes (e.g., Cytoplast^®^ TXT-200, Osteogenics Biomedical, Lubbock, TX, USA) are widely used and are considered the “gold standard”. Natural membranes are mostly represented by porcine or bovine collagen with products such as Bio-Gide^®^ (Geistlich, Wolhusen, Switzerland), EZ Cure^®^ (Biomatlante, Vigneux-de-Bretagne, France), or BioMend^®^ (Zimmer, Irvine, CA, USA). These materials do not cause any inflammatory responses but exhibit strong load bearing capacities against the compressive force of the soft tissue overlying them and provide adequate space for tissue regeneration [12,13,14]. Additionally, the clinical handling performed by doctors while applying membranes is crucial [15,16]. Due to their simplicity of use, quick formation of a solid membrane after injection, and ability to fill complex defects, injectable formulations are particularly appealing in this regard [13,17,18].

We previously reported silanization of hydroxypropyl methylcellulose by the addition of alkoxysilane groups [19,20,21], allowing formation of Si-HPMC hydrogels by silanol condensation. Numerous applications of this self-hardening hydrogel, including drug delivery for intervertebral discs, bone regeneration, and cartilage repair, have been thoroughly investigated [8,22,23,24]. Additionally, in a canine periodontal defect, we have shown that this cross-linked polymer can function as a physical barrier against cell invasion [24,25]. However, in the context of GBR, the main drawback of this self-curing hydrogel is its slow crosslinking process that hinders the rapid in situ formation of a membrane.

In situ light-curing formulations are already being used to prepare methacrylate resins to treat dental decay. This technique provides numerous advantages, such as spatiotemporal control over the crosslinking process [26,27,28,29]. Dextran methacrylate (DexMA) photopolymer, which undergoes crosslinking in the presence of a photoinitiator upon exposure to UV or visible irradiation, forms hydrogels that have been studied for multiple biomedical applications [30,31,32,33].

In the present study, we hypothesized that by combining Si-HPMC with DexMA, we could induce the formation of a novel IPN hydrogel. We also hypothesized that Si-HPMC/DexMA could be injected directly in situ during a GBR procedure and that a membrane would be rapidly obtained after irradiation with visible light. The synergistic effect obtained by the interpenetration of the two polymer networks led to improvement of the overall physicochemical properties, and the dense IPN network could provide a barrier effect against soft tissue invasion.

We synthesized DexMA polymers of various molecular weights, and we used riboflavin as a photoinitiator and a standard dentist lamp to cure the photo-crosslinkable polymer. At our knowledge, this is the first use of a photo-crosslinkable polymer as a regenerative membrane. This new biomaterial should be easy to handle and present a good biocompatibility during the healing phase. We analyzed the injectability of the IPN formulation as well as its biological and physicochemical properties. Finally, we compared the in-situ-obtained IPN hydrogel with a commercial membrane made of porcine collagen in a relevant GBR animal model. Calvaria critical-size defects (⌀ = 8 mm) in rabbits were filled with synthetic biphasic calcium phosphates granules in conjunction with the experimental membranes to assess their biocompatibility and efficacy in promoting bone growth. Quantitative analysis of the newly formed bone was performed using micro-computed tomography, and histological staining was used to visualize the soft tissue in the presence of the membranes.

## 2. Materials and Methods

### 2.1. Materials

Dextran 40, 100, and 500 kDa were purchased from Fluka (Milwaukee, WI, USA). Riboflavin 5′-phosphate sodium salt hydrate (RP), triethanolamine (TEOHA), glycidyl methacrylate (GMA), 4-(dimethylamino)pyridine (DMAP), dimethyl sulfoxide (DMSO), 4-(2-hydroxyethyl)piperazine-1-ethanesulfonic acid (HEPES), and neutral red (NR) were all obtained from Sigma-Aldrich (Saint Louis, MO, USA). Hydroxypropyl methylcellulose (HPMC) was provided by Colorcon^®^-Dow Chemical (Harleysville, PA, USA). Fetal bovine serum (FBS) was purchased from PAN-Biotech GmbH (Aidenbach, Germany). Penicillin, streptomycin, trypsin–EDTA, DMEM, and phosphate-buffered saline (PBS) were purchased from Gibco (Invitrogen, Carlsbad, CA, USA), and L929 cells were from the ATCC (Manassas, VA, USA). Biphasic calcium phosphate bone graft (MBCP+^®^) was purchased from Biomatlante (Vigneux de Bretagne, France) with the following characteristics: granules with a size of 0.5–1 mm and a 20/80 (*w*/*w*) ratio of HA/β-TCP (hydroxyapatite/β-tricalcium phosphate).

### 2.2. Dextran Methacrylate Synthesis

Synthesis of dextran derivatives was carried out according to the procedure described in the literature [30,32,34,35]. We prepared three methacrylate dextran derivatives by using dextran with molecular weights (Mw) of 40, 100, and 500 kDa (Table 1). In a typical synthesis, 20 g of dextran and 5 g of DMAP were solubilized in 220 mL of DMSO. In order to obtain dextran with a 20% methacrylation degree, 3.47 mL of GMA was added to the solution, which was kept in the dark for 48 h at room temperature under magnetic stirring. Finally, the reaction was stopped by adding 0.1 M HCl until a pH of 8 was reached. The solution was dialyzed (dialysis membrane cut-off 12–14 kDa, Spectrapor^®^), against distilled water, and then freeze dried. The degree of methacrylation of the final freeze-dried powder, defined as the number of methacrylated groups per 100 glucopyranose residues of dextran, was determined via ^1^H NMR in D_2_O using a Bruker AC-400 spectrometer.

### 2.3. Synthesis of Si-HPMC and Hydrogel Preparation

Si-HPMC was synthesized as previously described [21]. Briefly, HPMC was solubilized in a heptane/propanol/NaOH solution and grafted with 3-glycidoxypropyltrimethoxysilane (GPTMS). After this reaction was complete, the polymer was purified via dialysis and freeze-dried. HPMC was modified by the addition of silanol groups, with a degree of substitution of 0.6% (*w*/*w*), as determined using inductively coupled plasma-atomic emission spectroscopy [19,20,21]. Si-HPMC powder was dissolved in 0.1 M NaOH overnight and sterilized by autoclaving (121 °C for 20 min). Si-HPMC hydrogel was obtained by mixing the basic Si-HPMC solution with an acidic buffer in a 2:1 (polymer solution: acid buffer) ratio to reach a final pH of 7.4. The acidic buffer consisted of 0.06 M HCl, 1.8% (*w*/*v*) NaCl, and 6.2% (*w*/*v*) HEPES. The buffer was sterilized by autoclaving (121 °C for 20 min) prior to use.

### 2.4. Dextran Methacrylate Hydrogel Formation

We first determined the gelation and shape properties of the dextran derivatives as a function of the molecular weight, concentration, and photoinitiator concentration. Dextran methacrylate 40, 100, or 500 kDa was dissolved at concentrations of 5, 15, and 30% (*w*/*v*). One milliliter of each polymer solution was prepared in distilled water under magnetic stirring in a glass vial. A stock photoinitiator solution (PIS) composed of 4.2 mM riboflavin phosphate and 4.2 M TEOHA was prepared. HCl (6 M) was added to the solution to adjust the pH to balance the basicity of TEOHA to neutral pH [34]. The PIS was added to the dextran solutions at 1.5, 5, and 50 µL/mL (µL for each mL of the polymer solutions), and the mixtures were stirred for 5 min until homogeneous solutions formed. Then, the magnetic stirrer was removed, and the solutions were irradiated directly in the glass vial at a distance of 1 cm. Irradiation was performed from 30 to 240 s from one side, for 30 s each time, with a dentistry lamp (BA-Optima 10 curing light) at 1200 mW/cm^2^, 420–480 nm (B.A. International). Hydrogel formation was assessed by visual inspection (the vial tilting method) [36,37]. The gel point was reached when the solution/gel stopped flowing, and the gelation was considered completed when the entire hydrogel could be lifted with a spatula without any fluid flow. After gelling, the hydrogels were demolded and visually inspected. The capacity of the hydrogels to maintain the original shape was scored from ++ for very good, + for good, +/- for fair, - for poor, and -- for very poor. All further experimentations were then performed with Dex500MA20, referred to as DexMA.

### 2.5. DexMA Filter Sterilization

To sterilize the DexMA, a 30% (*w*/*v*) solution of Dex500MA20 in distilled water was passed through a 0.22 µm filter. The DexMA solution was filtered through an EMD Millipore Steriflip™ sterile disposable vacuum filter unit using a diaphragm vacuum pump N-022-AN.18 (KNF). The filtration was completed in approximately 15 min. The filtered solution was diluted (final concentration of approximately 5 mg/mL) and then freeze dried. The freeze-dried polymers were solubilized overnight (under gentle magnetic stirring) in an eluent composed of 0.1 M NaNO_3_ with 0.01% (*w*/*v*) NaN_3_ at a concentration of 1–2 mg/mL and then filtered before injection (0.22 µm). The solution was analyzed via a Viscotek TDA 305 Triple Detector GPC using three TSKgel^®^ GMPWXL (Tosoh) (100–1000 Å) columns with a flow rate of 0.6 mL/min. Dextran and DexMA were analyzed without filtration under vacuum for comparison.

To assess the stability of a concentrated polymer solution as a function of time, DexMA of 500 kDa was solubilized in distilled water to obtain a 30% (*w*/*v*) solution (stock solution). The stock solution was passed through a 0.22 µm filter and kept at 4 °C while avoiding exposure to light using an aluminum foil. The stock solution and the solution with the addition of methacrylic acid were analyzed at days 0, 1, 7, and 30. To detect the hydrolyzed groups, 15.225 ppm, 10.15 ppm, 5.075 ppm, or 2.03 ppm of methacrylic acid were added to some of the samples [38]. At different time points, 10 µL of the solutions were injected in an HPLC-UV (MERK with DAD) device equipped with a SunFire™ column (C18, 4.6 × 150 mm, 5 μm particles). The eluent was H_2_O/MeOH (60/40) with a flow rate of 1 mL/min.

### 2.6. IPN Hydrogel Preparation

Si-HPMC/DexMA solution was obtained by mixing the following three solutions: (1) Si-HPMC was solubilized in 0.1 M NaOH at 4% (*w*/*v*), (2) Dex500MA20 was solubilized at 30% (*w*/*v*) in distilled water with 5 µL of photoinitiator for each final mL of IPN solution. HCl (6M) was added to the solution to adjust the pH to balance the basicity of the TEOHA to neutral pH [34], and (3) acid buffer was prepared with 0.06 M HCl, 1.8% (*w*/*v*) NaCl, and 6.2% (*w*/*v*) HEPES. To obtain the Si-HPMC/DexMA solution, we mixed 1 volume of solution (1) with 2 volumes of solution (2) through two syringes connected with a Luer lock. The resulting solution was mixed with a 0.5 volume of solution (3) (Figure 1). The final solution contained 1.14% (*w*/*v*) Si-HPMC and 17.15% (*w*/*v*) DexMA (18.29% (*w*/*v*) of the total amount of polymer). The resulting Si-HPMC/DexMA solution was photo-crosslinked to obtain the IPN, as described for the DexMA hydrogel, with a standard dentist’s visible light lamp (B.A. International).

### 2.7. Rheological Analysis

The rheological measurements were performed with an ARES G2 rheometer (TA Instruments, New Castle, USA). We analyzed the Si-HPMC 1.14% (*w*/*v*), DexMA 17.15% (*w*/*v*), and Si-HPMC/DexMA (Si-HPMC 1.14% (*w*/*v*), DexMA 17.15% (*w*/*v*)) with or without irradiation. The data were collected using TRIOS 5.2 software. Time sweep experiments were performed (1% strain, 6.28 rad/s) using a parallel plate geometry (20 mm diameter) with an upper transparent plate of quartz that allowed material irradiation with a lamp to follow the gelation (λ = 450 nm, 430 mW, 350 mA, royal blue).

### 2.8. Injectability

The injectability of Si-HPMC 1.14% (*w*/*v*), DexMA 17.15% (*w*/*v*), and Si-HPMC/DexMA solution (Si-HPMC 1.14% (*w*/*v*), DexMA 17.15% (*w*/*v*)) were analyzed using a texture analyzer (TA DH plus) with a cylindrical aluminum probe of 25 mm, at 0.1 mm/s (compression rate) using a trigger force of 0.05 g. We ejected 0.5 mL of sample through a 3 mL syringe with an 18 G needle containing 1 mL of solution. The plots of force/displacement were recorded, and the mean force of injectability was taken from the plateau of the curve. The experiments were performed in triplicate and the results statistically analyzed using the Kruskal–Wallis test with GraphPad 6 software.

### 2.9. Hydrogel Apposition on BCP

To simulate the implant in the GBR model, the hydrogel membranes were cross-linked on top of BCP (granules with a size of 0.5–1 mm with 20% hydroxyapatite and 80% β-tricalcium phosphate (MBCP+^®^, Biomatlante, Vigneux de Bretagne, France). Cylindrical PLA molds (8 mm internal diameter and 1.5 mm height) were fabricated using a DiscoEasy 200 printer (Dagoma, Roubaix, France). The molds were filled with BCP and covered with freshly prepared polymer solutions. The solutions were injected (0.1 mL) through a syringe equipped with an 18 G needle onto the BCP. Si-HPMC/DexMA and DexMA solutions were photo-crosslinked for 120 s. A control group of Si-HPMC (not irradiated) was observed after 1 h of cross-linking. The samples were kept at room temperature prior to observation with an AxioZoom Macroscope (Zeiss).

### 2.10. Cytocompatibility

A neutral red (NR) uptake assay was carried out with L929 murine fibroblasts incubated with (1) DexMA extract or (2) PIS, Si-HPMC, DexMA, or Si-HPMC/DexMA solutions/hydrogels.

The cells were cultured in Dulbecco’s modified Eagle’s medium (DMEM) with 4.5 mg/mL glucose, 10% fetal bovine serum, and 1% penicillin–streptomycin. The cells were detached with 0.2% trypsin and seeded at 16,000 cells/mm^2^ in multiwell plates 24 h before each test.

To perform the NR uptake assay, an NR solution was prepared 24 h before the test (0.04 mg/mL in PBS) and incubated at 37 °C. Before addition to the cell culture medium, the NR solution was centrifuged to remove any crystals. The culture medium was discarded and 100 µL of NR solution was added to the cells and incubated for 3 h. After incubation, the NR solution was removed and a destaining solution (50% ethanol (96%), 49% deionized water, and 1% glacial acetic acid) was used to solubilize the dye trapped in the alive cells. The optical density was read at 550 nm using a microplate reader (Victor3V, PerkinElmer, Wellesley, MA, USA). The average optical density units were calculated after blank subtraction. All the tests were performed in triplicate. The results of the NR uptake assay are expressed as the ratio (%) between the absorbances of the experimental samples and the absorbance of the untreated cells.

To evaluate the cytocompatibility of the DexMA polymer extract, cells were seeded at 16,000 cells/mm^2^ in a 96-well plate for 24 h. Extracts of DexMA (Dex500MA20) in distilled water were prepared according to ISO 10993-5. A polymer solution of 30% (*w*/*v*) was placed in a dialysis tube (dialysis membrane cut-off 1 kDa, Spectrapor^®^). The tube was introduced into a Falcon tube with 10 mL of distilled water under stirring at room temperature for 72 h. The extract was sterilized with a 0.22 µm filter and used for testing the effect of leach-out product on cell viability. The culture medium was discarded from the cells and replaced with medium containing 10% (*v*/*v*) polymer extract. To examine the effect of irradiation (120 s), visible light from a lamp was used to expose cells in the wells with or without polymer extract. Any plates not exposed to visible light from the lamp were covered to prevent any uncontrolled exposure to light. The control wells contained cells with culture medium without polymer extract and were not irradiated, and they are referred to as the untreated cells. After 24 h of incubation, the medium was removed, the cells rinsed with sterile PBS, and neutral red solution was added to each well to perform the assay.

To evaluate the cytocompatibility in the presence of hydrogels, 48-well plates with inserts (polycarbonate, with a pore size of 8 µm) were used. The cells were seeded at 16,000 cells/mm^2^ in the wells, and 24 h later, Si-HPMC, DexMA, or Si-HPMC/DexMA solutions were added to the inserts. In order to examine the effect of irradiation, visible light curing was used to expose the cells and polymer solutions for 120 s to visible light from a lamp. In parallel, the cell viability in the presence of PIS was evaluated by adding the PIS to the culture medium at the same concentration as in the hydrogels. Any plates not exposed were carefully covered to prevent any uncontrolled light exposure. All the plates were returned to the 37 °C incubator for 24 h. The control wells contained cells without PIS and were not irradiated, and are referred to as the untreated cells. After 24 h, the medium was removed, the cells rinsed with sterile PBS, and neutral red solution was added to each well to perform the NR assay.

The experiments were performed in triplicate and the results statistically analyzed using the Kruskal–Wallis test with GraphPad 6 software.

### 2.11. In Vivo Experiments: Animal Model and Study Design

Studies were conducted on 12 male New Zealand White rabbits (3.39 ± 0.13 kg mean body weight) of 15 weeks of age. The rabbits were obtained from Granja San Bernardo (Navarra, Spain). All procedures were carried out under a project license (no. 010532/2018) approved by the National Competent Authority for animal research, called the Direção-Geral de Alimentação e Veterinária (DGAV, Lisbon, Portugal). The animals were housed in a purpose-designed room for the experimental animals, with one rabbit per cage, and were fed a standard laboratory diet. All experimental procedures were performed in accordance with European Directive 2010/63/EU and with National Law (DL no. 113/2013) on the protection of animals used for scientific purposes, and they were approved locally by the Animal Welfare Committee of the University of Trás-os-Montes e Alto Douro.

Four calvaria critical-size defects of 8 mm in diameter and full-thickness depth, according to the individual specimens, were surgically created in each animal. The defects were filled with BCP consisting of granules with a size of 0.5–1 mm with 20% hydroxyapatite and 80% β-tricalcium phosphate (MBCP+^®^, Biomatlante, Vigneux de Bretagne, France). The defects in 12 rabbits were filled with BCP and randomly treated with Si-HPMC, DexMA, Si-HPMC/DexMA, or collagen membrane as a control (EZ Cure^®^ Biomatlante, Vigneux de Bretagne, France). Once applied in situ, the DexMA and Si-HPMC/DexMA solutions were photo-crosslinked for 120 s. Owing to the adaptability of the membranes, no further stabilization using wound sutures was necessary. In order to confirm the inability of the 8 mm wounds to heal without treatment (critical size), we decided to substitute one BCP with the Si-HPMC hydrogel condition to add an empty defect to be analyzed 8 weeks after surgery. Bone repair and histological processing were analyzed at 2 weeks (*n* = 6) and 8 weeks (*n* = 6) after implantation.

### 2.12. Surgical Protocol

The animals were sedated via intramuscular injection of 1 mg/kg diazepam. They were anesthetized via intramuscular injection of 30 mg/kg ketamine and 5 mg/kg xylazine supplemented with intraoperative analgesia using 0.03 mg/kg buprenorphine. The animals were first shaved and disinfected to achieve aseptic surgical conditions. An incision was made along the middle of the cranium, and a full-thickness flap was performed to expose the calvaria. Once the calvaria was exposed, the entire flap was elevated, and four defects were then created using a trepan with a diameter of 8 mm under abundant saline irrigation. The defects were filled with BCP previously mixed with a few drops of sterile saline solution. The four previously described membrane conditions were randomly applied to the created defects. The flaps were then repositioned and closed with wound sutures. The postoperative management included subcutaneous administration of antibiotics (10 mg/kg enrofloxacin and 0.01 mg/kg buprenorphine). The surgery was carried out successfully. The rabbits were sacrificed 2 and 8 weeks later via intravenous administration of an overdose of sodium pentobarbital (>100 mg/kg).

### 2.13. Clinical Observation

The animals were carefully examined for inflammation, allergic reactions, or other complications around the surgical site throughout the entire healing period.

### 2.14. Micro-Computed Tomography Analysis

Qualitative analysis of newly formed bone was performed using a Bruker SkyScan-1272 high-resolution X-ray micro-computed tomography system (SkyScan 1272, Bruker, Kontich, Belgium). The following settings were used: resolution at 24 μm, 0.5 mm aluminum and 0.038 mm copper filter, 90 kV and 111 µA, rotation step 0.7°, and frame averaging of 3. Three-dimensional reconstructions were generated using NRcon^®^ software (Bruker) for each implant to assess bone formation in the control (BCP with collagen solid membrane) and the treated animals. The volume of interest (VOI) was defined for each defect, represented by the defect borders, and 76 sectional images were obtained in the coronal dimension. CTAn^®^ software (Bruker)was used to calculate the new bone formation volume and the BCP biomaterial volume. New bone formation was represented by the mineralized tissue subtracted from the BCP volume within the individual VOI. The lower grey value was set at 30 and the upper value at 255.

### 2.15. Statistical Analysis

The results of the cytocompatibility were analyzed using two-way ANOVA with the Bonferroni post hoc test. The results for the injectability or the new bone formation in micro-computed tomography were statistically analyzed using the Kruskal–Wallis test with GraphPad 6 software.

### 2.16. Histological Processing

The calvaria were separated from the cranium, and the specimens were fixed in 4% paraformaldehyde and then dehydrated via a graded series of ethanol treatments (70–100%). Non-decalcified bone specimens were infiltrated and embedded using Technovit^®^ 9100 (Heraeus Kulzer, Wehrheim, Germany). Blocks were cut into 7 µm slices with a microtome (SM2500, Leica, Frankfurt am Main, Germany) before staining. Only sections in the center of the defects were selected. The sections were stained with HES (hematoxylin, eosin, and safranin) and Goldner’s stain and then examined using a light microscope (Axioplan 2; Zeiss, Darmstadt, Germany).

## 3. Results

### 3.1. Polymer Synthesis

Si-HPMC was successfully modified by the addition of silanol groups with a 0.6% (*w*/*w*) degree of substitution as determined via inductively coupled plasma-atomic emission spectroscopy, as described elsewhere [19]. Si-HPMC 4% (*w*/*v*) sterile stock solution was prepared in NaOH as previously described [20,21].

Dextran methacrylate using polymers with Mw of 40, 100, and 500 kDa were synthesized. Table 2 summarizes the values of the theoretical and the measured degrees of methacrylation. NMR analysis indicated 18%, 17%, and 19% degrees of methacrylation, respectively, of methacrylate groups grafted along the polymer chains.

### 3.2. Dextran Methacrylate Hydrogel Formation

To select the starting dextran methacrylate solution to be combined with Si-HPMC, we determined the gelation and shape properties as a function of the dextran molecular weight, polymer concentration, and photoinitiator concentration. The sol-to-gel transition was characterized via the vial tilting method. The gel point was reached when the solution/gel stopped flowing, and the gelation was considered completed (gel time) when the entire hydrogel could be lifted without any fluid flow. After gelling, the demolded hydrogels were visually inspected at room temperature and their capacities to maintain their original shape were scored (Table 3).

We analyzed dextran methacrylate of 40, 100, and 500 kDa (Dex40MA20, Dex100MA20, and Dex500MA20) at 5, 15, and 30% (*w*/*v*) with 1.5, 5, or 50 µL/mL of PIS. We irradiated the solutions from 30 to 240 s for 30 s each time. With Dex40MA20, no hydrogel was obtained, irrespective of the polymer and the PIS concentration. Dex100MA20 formed hydrogels only at a high concentration (30% (*w*/*v*)) with 1.5 or 5 µL/mL of PIS. Dex500MA20 solutions formed hydrogels at every concentration tested with 1.5 or 5 µL/mL of PIS. Dex100MA20 and Dex500MA20 did not form hydrogels, irrespective of the concentration, even with 50 µL/mL of PIS. With Dex500MA20 15% and 30% (*w*/*v*), at the end of irradiation, the material was not homogeneous, with a cross-linked polymer layer on top of a liquid layer. On the other hand, Dex500MA20 with a low PIS concentration (1.5 µL/mL) formed hydrogels, but the gel point was delayed in time compared with 5 µL/mL of PIS.

Dex500MA20 was able to form a hydrogel with 5 µL/mL of PIS in 30 s. However, the hydrogels of Dex500MA20 at 30% and 15% (*w*/*v*) maintained their shape better after demolding compared with 5% (*w*/*v*). The polymer solution at 30% (*w*/*v*) with a molecular weight of 500 kDa and with 5 µL/mL of PIS was selected as a starting solution for IPN formulation due to the capacity to rapidly form a solid entity upon irradiation and for its ability to maintain a good shape. All further experimentations were performed with Dex500MA20, referred to from here onward as DexMA.

### 3.3. DexMA Filter Sterilization

To determine the impact of filtration of a 30% (*w*/*v*) solution of DexMA, the polymer was filtered and successively freeze-dried for analysis using HPLC-GPC. For comparison, dextran and unfiltered DexMA underwent the same procedure.

The addition of methacrylate groups along the dextran backbone increased the molecular weight of the polymer (Table 4). Filtration of DexMA also resulted in an increase in the molecular weight values compared with the unfiltered polymer. By contrast, the dispersity decreases for DexMA compared with dextran and for sterile DexMA compared with unfiltered DexMA.

### 3.4. DexMA Polymer Solution Stability

The stability of DexMA in aqueous solution was assessed since there is the possibility of methacrylic acid being released. The stability of the DexMA 30% (*w*/*v*) stock solution was analyzed on days 0, 1, 7, and 30. The solution was stored at 4 °C, protected from light. At each time point, the solution was analyzed via HPLC-UV to detect the presence of hydrolyzed groups. No methacrylic acid was detected in the samples for up to 30 days.

### 3.5. Rheological Analysis

The gelation kinetics of Si-HPMC at 1.14% (*w*/*v*), Si-HPMC/DexMA at 1.14% (*w*/*v*)/17.15% (*w*/*v*), and DexMA at 17.15% (*w*/*v*) were measured. The elastic and viscous moduli as a function of time are presented in Figure 2. The elastic and viscous moduli of Si-HPMC (Figure 2A) increased slowly, and the gel point was reached after 15 min (arrow) when the values of the moduli were approximately 10 Pa. After reaching the gel point, the elastic modulus gradually increased as a consequence of the crosslinking by silanol condensation. For Si-HPMC/DexMA (Figure 2B), analyzed without irradiation, the elastic and viscous moduli initially had similar values. The gel point was reached after 40 min (arrow), followed by a small increase in the elastic modulus.

DexMA solution (Figure 2C) was first irradiated with visible light from a lamp for 80 s. The curve shows an instantaneous gel point (arrow) and a sharp and immediate increase in the elastic and viscous moduli. After the exposure to light, the moduli values stabilized. During a second irradiation of 40 s, the elastic modulus increased to reach a value of 485 Pa, while the viscous modulus remained stable.

The Si-HPMC/DexMA initial elastic and viscous moduli values were an order of magnitude higher than those of each polymer solution separately. It is worth noting that while the non-irradiated gels had similar elastic and viscous modulus values, the irradiated DexMA and Si-HPMC/DexMA gels had higher elastic moduli of 800 Pa and 1000 Pa, respectively. When Si-HPMC/DexMA (Figure 2D) solution was exposed to 80 s of irradiation, the gel point was reached instantaneously and the elastic and viscous moduli increased rapidly. As observed for the DexMA solution, the crosslinking was dependent on the exposure to light from the lamp, and an increase in the elastic modulus to 1500 Pa was observed after a second irradiation of 40 s.

### 3.6. Injectability

The force necessary to inject a solution through a syringe with a needle was measured with a texture analyzer. For all of the solutions, the plots of force/displacement were recorded and the mean force of injectability was determined from the plateau of the curve. Si-HPMC, DexMA, and Si-HPMC/DexMA exhibited injectability force values of the same order of magnitude, between 1.4 and 1.6 N (Figure 3). All of the measured forces were under the limit of injectability specified in ISO 7886-1.

### 3.7. Hydrogel Apposition on BCP

To simulate the preparation of implants under GBR conditions, BCP was inserted in a mold and covered with Si-HPMC, DexMA, or Si-HPMC/DexMA solutions. The solutions were injected through a syringe equipped with an 18 G needle. The Si-HPMC/DexMA and DexMA solutions were photo-crosslinked for 120 s, and the Si-HPMC (not irradiated) was cross-linked for 1 h at room temperature. Si-HPMC is a transparent hydrogel and was thus barely visible above the BCP (not shown). Since its gel point is reached slowly after irradiation, polymer infiltration of the BCP was evident. After cross-linking, the biomaterial composed of Si-HPMC and BCP was analyzed. The most external part of the hydrogel could be manually removed from the BCP, and the remaining structure presented as a two-layer material of BCP and a cohesive interface composed of a dense surface of BCP embedded in Si-HPMC. Photo-crosslinked DexMA is a yellowish hydrogel (not shown), due to the presence of riboflavin (yellow color), and it was, therefore, easy to identify. The bilayer structure composed of the hydrogel and BCP was readily discernible. Upon observation of the interface of the hydrogel, after BCP removal, some BCP granules appeared to be trapped within it.

The IPN composed of Si-HPMC/DexMA polymers was transparent and translucent but also slightly yellowish due to the presence of riboflavin, as was present for the DexMA alone. The bilayer structure composed of IPN on BCP was readily identifiable. The photo-crosslinking process combined with a second viscous polymer system appeared to decrease the infiltration of the polymer solution in the BCP compared with DexMA or Si-HPMC. The BCP presented as a compact matrix with IPN. However, after IPN removal from the BCP layer, observation of the interface revealed that only a few granules were embedded in it (Figure 4).

### 3.8. Cytocompatibility

In order to evaluate the cytocompatibility of the polymers and the IPN, a neutral red uptake (NR) assay was carried out by incubating L929 murine fibroblasts with (1) DexMA extract and (2) PIS, Si-HPMC, DexMA, or Si-HPMC/DexMA solutions/hydrogels.

After synthesis of the DexMA polymer, potential contaminants were extracted to assess the cytocompatibility. The polymer extract, containing the leach-out products, was added to culture medium at 10% (*v*/*v*). The cytocompatibility results of cells exposed to polymer extract are reported in Figure 5A. There was no significant difference in the absorbance ratio of untreated cells compared to that of cells exposed to the polymer extracts. In addition, the cells irradiated for 120 s with visible light from a lamp did not exhibit a significant difference in the absorbance ratio compared to that of the untreated cells.

The cytocompatibility of cells was investigated in the presence of PIS, Si-HPMC, DexMA, or Si-HPMC/DexMA. In order to examine the effect of irradiation, visible light curing was used to irradiate the cells and PIS/polymer solutions for 120 s (Figure 5B). Irradiation, in presence of PIS, did not induce any difference in the absorbance ratio compared to that of the untreated cells. In addition, no significant differences were found between the absorbance ratio of cells cultured, whether or not irradiated, in the presence of Si-HPMC, DexMA, Si-HPMC/DexMA, or PIS.

### 3.9. Surgical Procedure and Clinical Observation

The surgery was successfully performed, with no postoperative complications (Figure 6). The animals were carefully examined for inflammation, allergic reactions, and other complications around the surgical site. None of the animals exhibited adverse reactions or complications relating to the presence of the implants throughout the entire healing period. No significant reductions in body weights were noted, and no postoperative infections were observed. All of the animals behaved normally during the healing phase.

### 3.10. Macroscopic Examination of Implants after Sacrifice

The animals were euthanized after two or eight weeks. The calvaria with periosteum were separated from the cranium. Macroscopic examination was performed prior to micro-computed tomography analysis and histological processing. Most of the samples exhibited a regular morphology and color. Only one defect condition presented formation of a clot, which corresponded to the empty defect condition. The periosteum appeared uniform in every sample and was adherent to the calvaria. None of the samples presented a perforation or membrane/BCP exposure. No BCP granules were found around the defects, and the membranes were not visible at the internal or the external side of the calvaria two weeks after the surgery. At 8 weeks after the surgery, some BCP granules were visible around the defects, but this was not related to a specific condition.

### 3.11. Micro-Computed Tomography Analysis

Images and quantification of calvaria defect healing were studied at 2 and 8 weeks after the surgery. Representative pictures (coronal and axial planes) of implants of BCP covered with a commercial collagen solid membrane (control), Si-HPMC, DexMA, or IPN hydrogel membranes, as well as quantitative analysis of the mineral tissues, are presented for 2 (Figure 7) and 8 weeks (Figure 8) after the surgery. We decided to add an empty defect to be analyzed at 8 weeks after the surgery to validate the inability of the 8 mm wound to heal without treatment.

Figure 7A shows a representative picture of the collagen, Si-HPMC, DexMA, and IPN defects 2 weeks after the surgery. The borders of all defect groups could be identified in each condition. All of the defects appeared to be filled with mineral bone graft, and the amount of BCP appeared to be homogeneous in each defect, with no discernible empty spaces. Axial images confirmed the homogeneous BCP distribution in the defect volume. No qualitative differences were noticed between the various conditions. Quantitative analysis of the total mineralized content, new bone, and BCP was performed in the volume of interest using CTAn^®^ software (Figure 7C). The quantification was expressed as a percentage of the mineral volume (MV) normalized by the total volume (TV) of the defect. The results show a homogeneous BCP distribution in the experimental membranes comparable to that of the control group of approximately 15%. The amount of new bone was 27.01 ± 7% for collagen, 33.67 ± 4.60% for Si-HPMC, 33.60 ± 11.8% for DexMA, and 27.71 ± 14% for IPN. The amount of new mineral tissue deposited did not differ from that of the collagen membrane control material. However, none of the defects had completely recovered after 2 weeks, and the mineralized volume of the material represented approximately 40% of the total volume, indicating that the healing process was not completed.

Images and quantification of calvaria defect healing 8 weeks after the surgery are presented in Figure 8. In order to confirm the critical size of the defects created in the calvaria, we added an empty condition (no BCP, no membrane), which was also analyzed 8 weeks after the surgery. The coronal and axial views presented in Figure 8 show an empty defect with no mineralization inside the volume of interest. This confirms the inability of the empty defect to heal in 8 weeks. Collagen, Si-HPMC, DexMA, and IPN conditions after 8 weeks from surgery presented in most of sample empty spaces (no BCP, no bone). These spaces do not appear to be related to a particular condition or a specific position of the defect in the calvaria. Quantitative analysis revealed a total mineral tissue of approximately 44% and a BCP distribution of 11%. The amount of new bone at 8 weeks after the surgery was 34.30 ± 8.17% for collagen, 33.74 ± 9.94% for Si-HPMC, 35.19 ± 6.74% for DexMA, and 30.49 ± 5.23% for IPN, with no significant differences between the groups.

### 3.12. Histological Analysis

After sacrifice, the skull containing all four craniotomy sites was removed and sectioned. Histological analysis found new bone formation in all of the implanted groups as well as in the positive control, but only limited new bone formation was observed in the control empty defects (Figure 9). The various membranes (collagen, Si-HPMC, DexMA, and IPN) were still identifiable 2 weeks after the surgery (Figure 10) but not at 8 weeks. No signs of acute inflammation were visible around the membranes (Figure 10). The quality of the new bone formation in the membrane groups was similar to that observed in the positive control (Figure 10 and Figure 11).

## 4. Discussion

Interpenetrating polymer networks (IPNs) are systems composed of two or more polymer networks interwoven at the molecular level, but not linked to each other. Simultaneously cross-linked in situ IPNs have been used in minimally invasive surgery, and they may be associated with reduced postoperative morbidity and pain [39].

In the present study, we hypothesized that by combining a chemically cross-linked Si-HPMC hydrogel with a photo-crosslinked DexMA entity, we could induce the formation of an innovative IPN hydrogel. A Si-HPMC/DexMA solution could be injected directly in order to rapidly obtain a membrane for GBR by visible light irradiation.

Si-HPMC polymer was successfully synthesized, and a sterile polymer solution was prepared, as previously described [19,20]. Three different dextran methacrylate derivatives of 40, 100, and 500 kDa with theoretical methacrylation degrees of 20% were prepared. The NMR results revealed a percentage of methacrylation in agreement with the theoretical values.

To photo-crosslink the methacrylate polymer, we selected a riboflavin-based photoinitiator, used with triethanolamine as a co-initiator [40,41]. To improve the relevance of the process to current dental practice, we used a lamp emitting visible light (λ 420–480 nm) rather than a UV lamp to cross-link the solution [42,43,44]. Specifically, we chose a standard dentistry lamp, which is a very common instrument in current dental clinical practice. To select the starting dextran methacrylate solution to be combined with Si-HPMC, we determined the gelation and shape properties as a function of the DexMA molecular weight, concentration, and photoinitiator concentration. An inverse proportionality was noted between the polymer concentration and the time required to reach the gel point. Indeed, probably due to the increased presence of methacrylate moieties, the concentrated solutions had reduced times for reaching the gel point. On the other hand, increasing the polymer molecular weight also reduced the time to reach the gel point. Conversely, DexMA with the lowest molecular weight (40 kDa) was not able to form a hydrogel membrane within the observed irradiation time. Initiation of the crosslinking process requires a precise and adequate photoinitiator concentration. Indeed, no hydrogel formation was observed when the photoinitiator concentration was increased to 50 µL/mL. This could be attributed to the increased opacity of the hydrogel precursor solution when the photoinitiator content is increased, which could hinder the penetration of visible light into the precursor solution. On the other hand, DexMA formed hydrogels at a low photoinitiator concentration (1.5 µL/mL), although the gel point was delayed compared to 5 µL/mL. Based on these results, we selected 500 kDa DexMA at 30% (*w*/*v*) with 5 µL/mL of photoinitiator as the starting solution for making IPNs.

Chu et al. described a DexMA cross-linked in the presence of different concentrations of riboflavin with L-arginine as the co-initiator. The study describes a solution of 25% by weight of dextran (64–75 kDa) that was weakly substituted with methacrylic anhydride (% MA 0.287). The solution was photo-crosslinked with a fluorescent lamp for 40 min. They found that only a low concentration of riboflavin was necessary to cross-link the polymer (0.01–0.5%) and that increasing its concentration did not lead to a cross-linked biomaterial, which is in line with our observations in the present study [45]. Menzel investigated the photo-crosslinking of a dextran modified with hydroxyethyl methacrylate groups with a visible light lamp and using camphorquinone as a photoinitiator with various types of co-initiators. Hydrogel formation occurred after 120 s of irradiation, using a 35–45 kDa polymer (concentration 10–30%) and by placing the light source directly over the polymer solution. However, for this study, due to the poor solubility of camphorquinone, the cross-linking was performed in the presence of DMSO [43]. In our study, we chose riboflavin, and in particular its phosphate derivative, as a photoinitiator, due to its water solubility, thereby avoiding the use of organic solvent in the final formulation.

The selected DexMA solution was sterilized by filtration through a 0.22 µm filter, and the polymer molecular weight before and after filtration was analyzed using GPC. During the filtration process, exposure of DexMA to light could not be avoided. Due to the high reactivity of methacrylic groups, even in the absence of the photoinitiator, the filtration process led to an increase in the molecular weight of the filtered DexMA compared with the unfiltered and, therefore, not light-exposed polymer. This is in agreement with the observed reduction in the molecular weight dispersity of the polymer after filtration.

The IPN was obtained by combining the two preformed solutions to obtain a final polymer solution containing 1.14% (*w*/*v*) Si-HPMC and 17.15% (*w*/*v*) DexMA. IPN multicomponent hydrogels are based on the ability of each polymer to form an independent network in the presence of the other polymer, which—in the most favorable case—leads to the synergistic combination of the properties of each polymer network. Rheological analysis of the combined Si-HPMC/DexMA solution confirmed crosslinking of Si-HPMC in the presence of DexMA. The DexMA polymer was also cross-linked in the presence of Si-HPMC, with no modification of the kinetics of the photo-crosslinking. This is likely due to the low number of crosslinking nodes in Si-HPMC after mixing, which facilitates diffusion and reaction of radical species. The initial values of the elastic and viscous moduli of Si-HPMC/DexMA were an order of magnitude higher than those of each polymer solution separately, and after irradiation, an elastic modulus of 1500 Pa was achieved. Moreover, as shown in Figure 2, with IPN and DexMA hydrogels, immediately after photo-crosslinking, two strong gels were obtained (G′ >> G″) compared to Si-HPMC, in which, after the crossover (G′ > G″), the elastic modulus increased progressively, albeit slowly, at this concentration.

The injectability of the IPN solution before crosslinking was investigated using a 3 mL syringe with an 18 G needle. Our data show that the solution combining Si-HPMC with DexMA remained injectable, as for each polymer solution. The force required to inject the combined solution was below the recommended limit, thus making it suitable for in situ applications. Injectable formulations capable of filling complex defects and rapidly forming solid membranes upon injection are particularly attractive because of their ease of handling.

The synergistic effect obtained by the interpenetration of the two polymer networks improved the physicochemical properties, with the dense network ensuring a barrier effect against soft tissue invasion and delaying the hydrogel resorption. Most resorbable membranes used to date, made of polylactic acid or animal-derived collagen, are characterized by rapid resorption kinetics after implantation. Thus, once the membranes are degraded, they no longer support the underlying regenerating process. In order to improve their properties, commercially available membranes can also be made of cross-linked polymers. Indeed, the presence of physical or chemical crosslinking nodes delays the loss of their barrier properties [45]. Commercially available cross-linked membranes can thus delay resorption. For example, BioMend^®^ is composed of linear type I bovine collagen and degrades in 6–8 weeks, whereas BioMend Extend^®^, with a similar composition but cross-linked with formaldehyde, degrades in 18 weeks [46]. In a systematic review, Marquez et al. reported a comparison between the clinical outcomes of cross-linked and linear resorbable collagen membranes in terms of the regenerated bone volume and postoperative complications during bone regeneration procedures. The bone growth results were comparable, whereas the postoperative complications suggested better outcomes with cross-linked collagen membrane 4 to 6 months after the surgery [45,47].

Hydrogels are hydrophilic three-dimensional networks that allowthe diffusion of nutrients, thus ensuring cell viability. In previous experiments, we used Si-HPMC as a self-hardening hydrogel for cell encapsulation. The in vitro results confirmed the ability of Si-HPMC to act as a physical barrier, trapping cells inside the hydrogel. In addition, cell viability was demonstrated up to 21 days, and diffusion experiments confirmed nutrient diffusion, which is essential for cell survival [8,48,49,50]. This physical barrier property of Si-HPMC hydrogel has also been confirmed in several in vivo studies [24,25,50]. Si-HPMC has also been reported to form an IPN with calcium–alginate polymer. The addition of the calcium–alginate network increased the stiffness of Si-HPMC, while maintaining the cell viability of the SW1353 cell line in 2D or encapsulated inside the hydrogel up to 7 days [51]. Dextran hydroxyethyl methacrylate has been described in an IPN with calcium–alginate. The polymer mixture instantaneously increased the mechanical properties under UV irradiation and ensured survival of the chondrocytes encapsulated in the IPN hydrogel [35].

In this study, the cytotoxicity was evaluated using a neutral red assay of murine fibroblasts according to the ISO 1993-5 standard. The neutral red uptake test was performed on cells in the presence of polymer extract and in the presence of polymer solutions/hydrogels. These cell viability tests did not reveal a decrease in neutral red uptake compared with the untreated cells. In addition, no cytotoxic effect was noted in the presence of the photoinitiator or after irradiation.

Based on these encouraging in vitro results, we performed an in vivo study to assess the efficacy and biocompatibility of our injectable in situ hydrogel membrane in conjunction with BCP granules. We used a commercial porcine collagen membrane as a positive control. Use of a membrane in the GBR process in calvaria defects is of paramount importance. The presence of a membrane promotes new bone formation by preventing soft tissue infiltration and by stabilizing BCP granules during surgical manipulation [52]. Rabbit calvaria defects, when treated only with BCP granules, had a low percentage of new bone regeneration of approximately 10% after 21 days compared with other anatomical defects such as the femur (50%) or the tibia (20%) [53]. Hawake et al. treated defects of 8 mm in rabbit calvaria with BCP granules. After six weeks, quantitative analysis found 17% new bone formation [54]. Another study has described calvaria defects in rats treated with human bone morphogenetic protein-2 with and without the addition of a collagen membrane. New bone formation was significantly higher with a collagen membrane (46%) compared with that without (26%) [55].

Throughout the entire healing period, none of the animals exhibited adverse reactions or complications related to the presence of the implants. Micro-CT analysis confirmed the presence and equal distribution of BCP in the defects treated with Si-HPMC, DexMA, and IPN compared to collagen membrane. Quantitative assessment of the mineral deposition found a mineral deposition in Si-HPMC, DexMA, and IPN comparable to the collagen membrane condition of approximately 33.43% after 8 weeks of healing. Histologically, the Si-HPMC/DexMA and IPN membranes were still present after 2 weeks of healing. However, after 8 weeks, the membranes were no longer detectable, but a membrane effect is suggested in all conditions in light of the absence of invagination of the soft tissues between the BCP granules. The new bone formation was readily apparent and statistically significant compared with the spontaneous healing in the negative control group. The collagen and hydrogel membranes were also not visible in micro-CT analysis after 8 weeks of healing. This new, innovative, photo-crosslinked IPN membrane was easy to handle during the surgical procedure. Use of the IPN injectable hydrogel membrane for GBR appears to be as efficient as the reference positive control collagen membrane, with no adverse inflammatory reaction, complete degradation after 8 weeks, and similar new bone formation inside the calvarial defects.

## 5. Conclusions

In this work, we devised a novel IPN hydrogel membrane composed of a self-setting Si-HPMC and photo-crosslinkable DexMA. The Si-HPMC/DexMA solution was injected directly into bone defects, and an IPN hydrogel was rapidly obtained in situ by irradiation with visible light. Throughout the healing period, none of the animals exhibited any adverse reactions or complications due to the presence of the membranes. The efficacy of the resulting IPN membrane was compared in a relevant animal model of GBR with that of a commercial porcine collagen membrane. The injectable membrane exhibited extremely interesting properties, including its ease of handling, in addition to preventing material flow during the time between application and gelation and its ability to cover the defect filled with BCP granules. Bone tissue regeneration was comparable in the defects treated with Si-HPMC, DexMA, or IPN and the conventional product used in clinical practice. In conclusion, we demonstrate that this innovative IPN hydrogel, which is easy to handle and apply, constitutes an excellent non-animal alternative to porcine collagen for bone healing applications. However, complementary preclinical studies with larger animal models, followed by clinical studies, are necessary to definitively validate this new, innovative, photo-crosslinked IPN membrane.

## Figures and Tables

**Figure 1 bioengineering-10-00094-f001:**
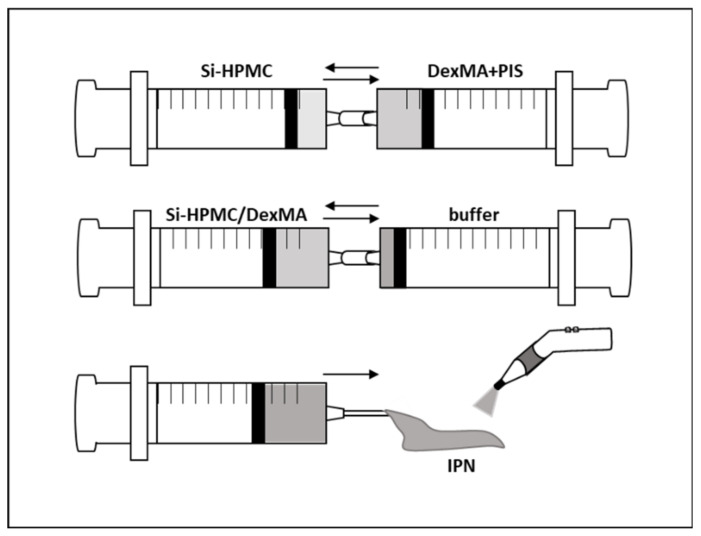
Preparation and photo-crosslinking of IPN. First, solutions of Si-HPMC and DexMA containing the PIS were mixed together using two syringes connected with a Luer lock. The acidic buffer was then added to the mixture using another syringe connected with a Luer lock. The IPN solution was then ready to be injected and photo-crosslinked (420–480 nm). Si-HPMC: silanized hydroxypropyl methylcellulose, DexMA: dextran methacrylate, PIS: photoinitiator solution, IPN: interpenetrating polymer network.

**Figure 2 bioengineering-10-00094-f002:**
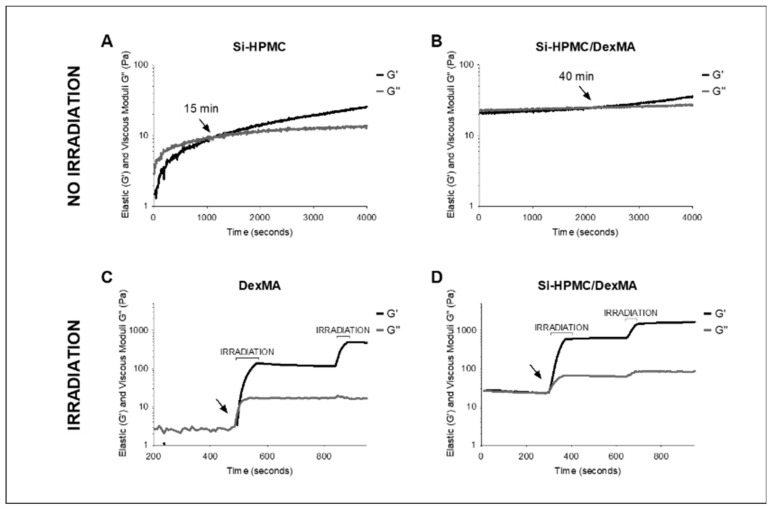
Elastic (G′) and viscous moduli (G″) as a function of time (seconds) of (**A**) Si-HPMC and (**B**) Si-HPMC/DexMA without irradiation, (**C**) DexMA, and (**D**) Si-HPMC/DexMA under a first irradiation of 80 s and a second irradiation of 40 s. The gel point is indicated by an arrow. The polymer concentrations were Si-HPMC at 1.14% (*w*/*v*), Si-HPMC/DexMA at 1.14% (*w*/*v*)/17.15% (*w*/*v*), and DexMA at 17.15% (*w*/*v*). The data were recorded at 6.8 rad/s frequency, 1% strain, and room temperature, and the lamp used for irradiation had λ = 450 nm. Si-HPMC: silanized hydroxypropyl methylcellulose, DexMA: dextran methacrylate, Si-HPMC/DexMA: silanized hydroxypropyl methylcellulose/dextran methacrylate.

**Figure 3 bioengineering-10-00094-f003:**
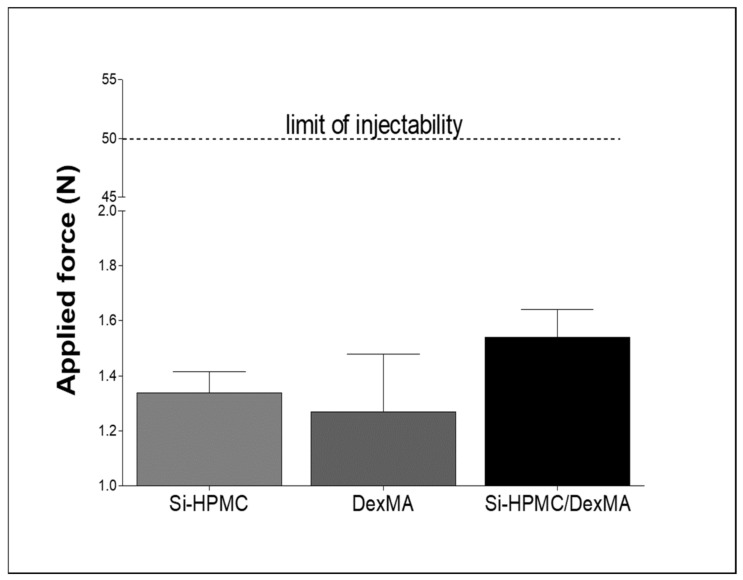
Injectability of Si-HPMC, DexMA, and Si-HPMC/DexMA solutions. The force required to inject 1 mL of each polymer solution through a 3 mL syringe loaded with 1 mL of polymer solution using an 18 G needle. The results are expressed as means ± SEM (*n* = 3). Si-HPMC: silanized hydroxypropyl methylcellulose, DexMA: dextran methacrylate, Si-HPMC/DexMA: silanized hydroxypropyl methylcellulose/dextran methacrylate.

**Figure 4 bioengineering-10-00094-f004:**
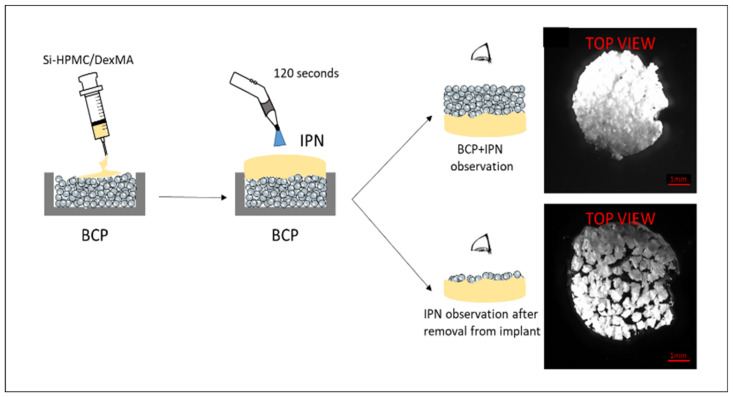
Schematic outline of the preparation of an implant composed of BCP and IPN hydrogel membrane. The cylindrical mold was filled with BCP. Si-HPMC/DexMA solution was injected onto the BCP surface. The solution was photo-crosslinked for 120 s with visible light from a lamp to form the IPN. BCP with IPN and IPN after removal from the BCP were observed with a microscope. Mold with an 8 mm diameter and a 1.5 mm height. Si-HPMC: silanized hydroxypropyl methylcellulose, DexMA: dextran methacrylate, Si-HPMC/DexMA: silanized hydroxypropyl methylcellulose/dextran methacrylate, BCP: biphasic calcium phosphate, IPN: interpenetrating polymer network.

**Figure 5 bioengineering-10-00094-f005:**
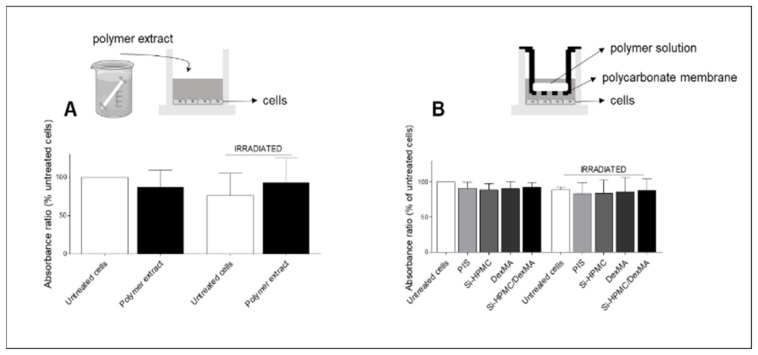
Cytocompatibility assessed via neutral red uptake by L929 cells seeded at 16,000/cm^2^ after (**A**) 24 h in the presence of DexMA polymer extract, (**B**) 24 h in the presence of 5 µL/mL of PIS and Si-HPMC, DexMA, or Si-HPMC/DexMA. In order to examine the effect of irradiation, the solutions were exposed to visible light from a lamp for 120 s. The results are presented as means ± SEM (*n* = 3) and analyzed using two-way ANOVA with the Bonferroni post hoc test. Si-HPMC: silanized hydroxypropyl methylcellulose, DexMA: dextran methacrylate, Si-HPMC/DexMA: silanized hydroxypropyl methylcellulose/dextran methacrylate, PIS: photoinitiator solution.

**Figure 6 bioengineering-10-00094-f006:**
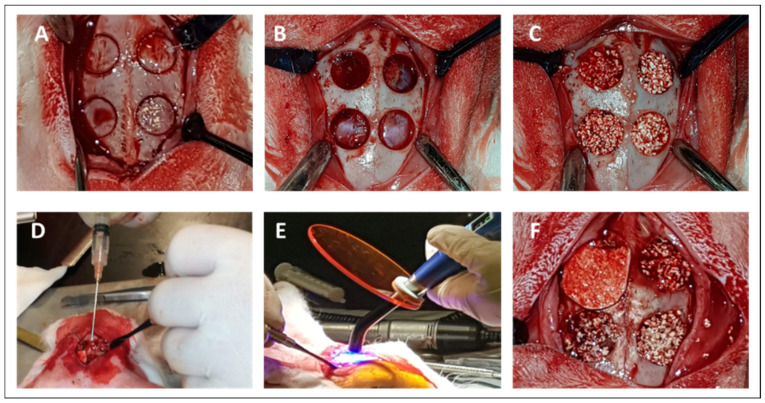
Surgical procedure on rabbit calvaria. (**A**) Surgical calvaria incisions. (**B**) Four defects (Ø = 8 mm) were created in each calvaria of 12 male New Zealand White rabbits, and (**C**) the defects were filled with BCP. (**D**) Si-HPMC/DexMA application using a syringe with an 18 G needle, (**E**) Si-HPMC/DexMA photo-crosslinking with a standard dentistry lamp for 120 s to form the IPN membrane. (**F**) Random assignment of membrane apposition on the defect, clockwise from top left: control collagen membrane, Si-HPMC, IPN, and DexMA. Si-HPMC: silanized hydroxypropyl methylcellulose, DexMA: dextran methacrylate, Si-HPMC/DexMA: silanized hydroxypropyl methylcellulose/dextran methacrylate, IPN: interpenetrating polymer network.

**Figure 7 bioengineering-10-00094-f007:**
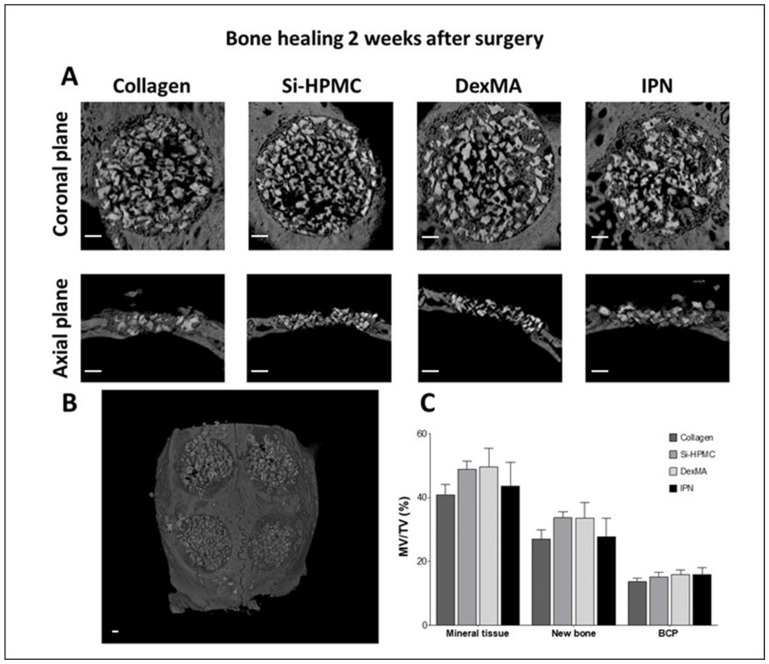
Bone healing evaluation 2 weeks after implantation (**A**) Micro-CT imaging of the calvaria defects with diameters of 8 mm (**B**) 3D reconstruction of the external view of calvaria. Clockwise from top left: control collagen membrane, IPN, DexMA, and Si-HPMC. (**C**) Quantification of the percentage of mineral volume versus the total volume (MV/TV) of the total mineral tissue, new bone, and biphasic calcium phosphate granules (BCP) in the volume of interest. Error bars represent the standard deviations of the means (*n* = 6). Si-HPMC: silanized hydroxypropyl methylcellulose, DexMA: dextran methacrylate, IPN: interpenetrating polymer network. No statistical difference was found between the groups. (Scale bar: 1 mm).

**Figure 8 bioengineering-10-00094-f008:**
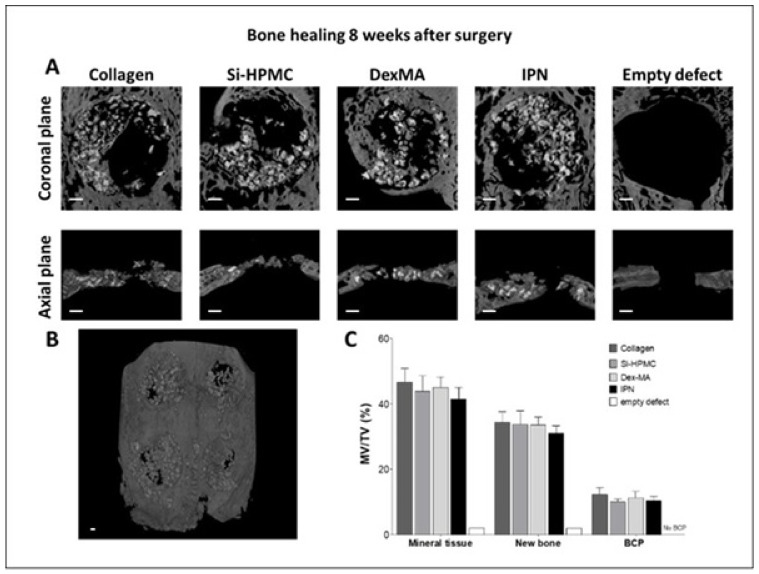
Bone healing evaluation 8 weeks after implantation (**A**) Micro-CT imaging of the calvaria defects with diameters of 8 mm, (**B**) 3D reconstruction of the external view of the calvaria. Clockwise from top left: control collagen membrane, Si-HPMC, IPN, and DexMA. (**C**) Quantification of the percentage of mineral volume versus the total volume (MV/TV) of the total mineral tissue, new bone formation, and biphasic calcium phosphate granules (BCP) in the volume of interest. Error bars represent the standard deviations of the means, collagen *n* = 6, Si-HPMC *n* = 5, DexMA *n* = 6, IPN *n* = 6, empty defect *n* = 1. Si-HPMC: silanized hydroxypropyl methylcellulose, DexMA: dextran methacrylate, IPN: interpenetrating polymer network. No statistical difference was found between the groups. (Scale bar: 1 mm).

**Figure 9 bioengineering-10-00094-f009:**
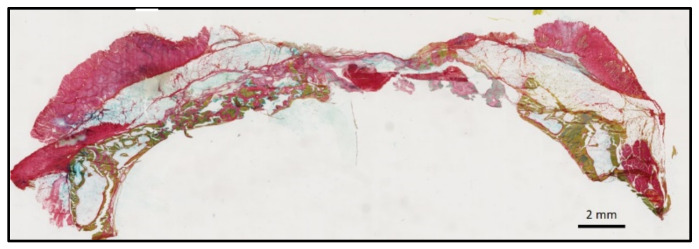
Histologic view of calvaria defects at 8 weeks showing—(**left**) the IPN group, with the defect mainly occupied by the bone graft material and the newly formed bone, and (**right**) the control empty group, with the defect mainly occupied by soft tissues, which collapsed inside the defect, and limited new bone formation at the defect boundaries.

**Figure 10 bioengineering-10-00094-f010:**
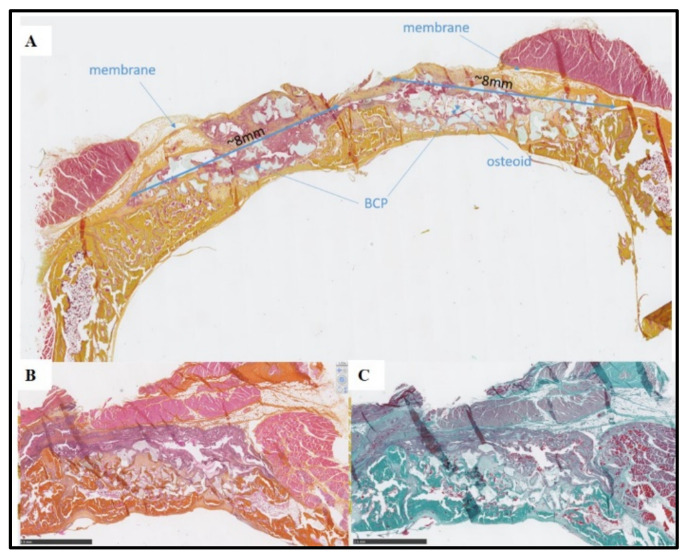
Histological views at 2 weeks. (**A**) Overall view of the calvaria with the two defects filled by BCP granules and Si-HPMC membrane on the left and DexMA membrane on the right. (**B**,**C**) Higher magnification of the defect implanted with BCP granules and covered with the IPN membrane, with HES (**B**) and Goldner (**C**) staining. After 2 weeks, there were no signs of excessive inflammation in the soft tissue, (**B**) the defects were filled by the BCP granules, and osteoid tissue can be seen around the granules (**C**). The IPN membrane can be clearly identified up to the bone graft material (**B**,**C**).

**Figure 11 bioengineering-10-00094-f011:**
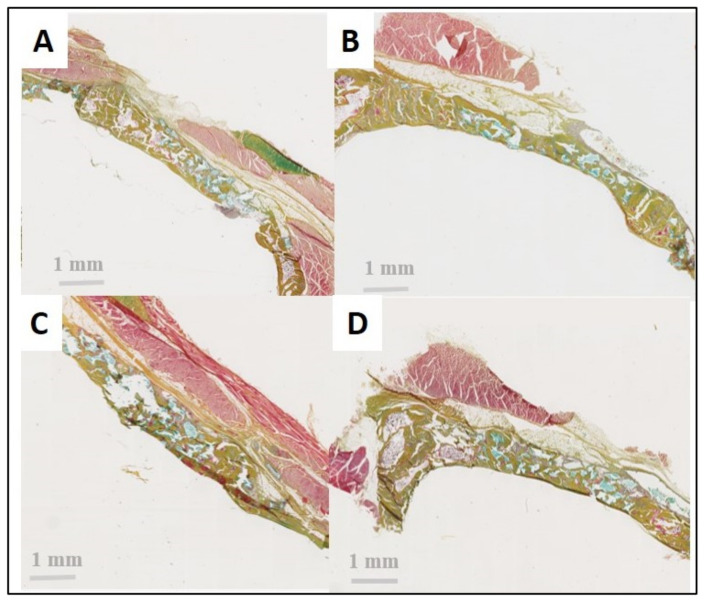
Histological views after 8 weeks of healing (Movat staining). (**A**) Positive control with collagen membrane, (**B**) Si-HPMC, (**C**) DexMA, and (**D**) IPN. In all groups and in the positive control, the delineation between the BCP granules and the soft tissue is readily apparent. There is no invagination of the soft tissue, thus confirming the barrier effect of the membranes. The new bone formation appears more homogeneous in the IPN and the positive control compared with the Si-HPMC and DexMA groups.

**Table 1 bioengineering-10-00094-t001:** The polymers and materials used in this study.

Abbreviation	Full Name
**Si-HPMC**	Silanized hydroxypropyl methylcellulose
**Dex40MA20**	Dextran 40 kDa with a theoretical methacrylation degree of 20% *
**Dex100MA20**	Dextran 100 kDa with a theoretical methacrylation degree of 20% *
**Dex500MA20**	Dextran 500 kDa with a theoretical methacrylation degree of 20% *
**PIS**	Photoinitiator solution
**IPN**	Interpenetrating polymer network
**BCP**	Biphasic calcium phosphate

* The methacrylation degree is defined as the number of methacrylated groups per 100 glucopyranose residues of dextran.

**Table 2 bioengineering-10-00094-t002:** Theoretical and experimental degree of methacrylation (% MA) of Dex40MA20, Dex100MA20, and Dex500MA20. The degree of methacrylation, defined as the number of methacrylated groups per 100 glucopyranose residues of dextran, was determined via proton NMR at 300 MHz. Dex40MA20: Dextran 40 kDa with a theoretical methacrylation degree of 20%, Dex100MA20: Dextran 100 kDa with a theoretical methacrylation degree of 20%, Dex500MA20: Dextran 500 kDa with a theoretical methacrylation degree of 20%.

	Theoretical Values	Experimental Data
Polymer	% MA	% MA
**Dex40MA20**	20	18
**Dex100MA20**	20	17
**Dex500MA20**	20	19

**Table 3 bioengineering-10-00094-t003:** Gelation and shape properties of dextran methacrylate were analyzed. Dex100MA20 and Dex500MA20 at 5, 15, and 30% (*w*/*v*) with 1.5 or 5 µL/mL of photoinitiator solution. The sol-to-gel transition was characterized using the vial tilting method. The gel point was reached when the solution/gel stopped flowing, and the gelation was considered completed (gel time) when the entire hydrogel could be lifted without any fluid flow. After gelling, the demolded hydrogels were visually inspected and the capacity to maintain the original shape was scored from ++ for very good, + for good, +/- for fair, - for bad, and -- for very bad. The symbol/indicates that no hydrogel formation occurred. Dex100MA20: Dextran 100 kDa with a theoretical methacrylation degree of 20%, Dex500MA20: Dextran 500 kDa with a theoretical methacrylation degree of 20%. * No Dex40MA20 hydrogel was obtained, and the gelation and shape properties could not be determined.

		Photoinitiator Concentration (µL/mL)					
		**1.5**			**5**		
**Polymer ***	**Conc. % (*w*/*v*)**	**Gel Point** **(s)**	**Gel Time** **(s)**	**Shape**	**Gel Point** **(s)**	**Gel Time** **(s)**	**Shape**
**Dex100MA20**	**5**	/	/	/	/	/	/
	**15**	/	/	/	/	/	/
	**30**	150	210	+	90	150	-
**Dex500MA20**	**5**	210	240	--	30	150	--
	**15**	90	150	--	30	120	+
	**30**	60	60	++	30	90	++

**Table 4 bioengineering-10-00094-t004:** Molecular weights and molecular weight distributions of dextran, DexMA, and DexMA passed through a 0.22 µm filter. The polymer solutions were analyzed via GPC calibrated using dextran standards. Dextran with a theoretical methacrylation degree of 20%. DexMA: dextran methacrylate, Ɖ: dispersity (Mw/Mn).

	Dextran	DexMA	Filtered DexMA
**M_w_ (kDa)**	500	702.8	803.4
**M_n_ (kDa)**	52.9	144.3	171.5
**Ɖ**	9.45	4.87	4.68

## Data Availability

The data that support the findings of this study are available from the corresponding author upon reasonable request.

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
