# Peer review of "Injectable Hydrogel Membrane for Guided Bone Regeneration"

_bioengineering, 2023, doi:10.3390/bioengineering10010094_

Round 1

Reviewer 1 Report

We would like to first congratulate the authors for their research and their sustained effort put into this article.

This article proposes a new type of membrane which may have future clinical uses into guided bone regeneration. The extensive research protocol which includes in vivo and in intro analysis for the purpose of characterizing the membrane.

I must remark the fact that the authors have chosen a rabbit calvaria model with 14 specimens with a critical size defect of 8mm. The use of BCP as a filler material can be discussed in other studies, it has its advantages and its drawbacks. The fact that on the same calvaria there were 4 defects could be a risk of contaminating the samples and influencing the defect healing. The fact that the defects were randomly assigned to one group adds a plus to the article.

The use itself of the calvaria model, although vey common and easy, it has its drawbacks since the bone in cancellous and it does not have the same functional load and the same healing potential as alveolar bone which is the most common subject of GBR.

I am strongly recommending the publication of this paper in its current form because the authors have managed to gather all the necessary information for the reader to make use and understand the material which was tested and the results presented.

Author Response

We sincerely thank this reviewer for his very positive review of our article.

Reviewer 2 Report

This manuscript proficiently discussed a novel injectable composite hydrogel-forming membrane for GBR models both in vitro and in vivo. The design of the experiments was clear and well-elaborated. The reviewer was impressed by the proficiency of the work, displaying in vivo functionalities in rabbit GBR, which could lead to larger preclinical animal models in the future.

However, there are some issues the authors could have addressed before accepting for publication.

The reviewer considered the writing as not concise. In the abstract, the novelty of the research should be clearly highlighted. The existing research work covered in the discussion part should be short and concise, as it is where your significance was to be discussed and highlighted. 

Figure 1,4&5 could be combined as one schematic diagram showing the preparation and characterization of the INP.

The solution of almost every figure was low, and the font size of texts and legends must be uniform.

The scale bars must be given in Figure 7&8.

P values must be calculated among groups in Figure 7&8 C

Author Response

This manuscript proficiently discussed a novel injectable composite hydrogel-forming membrane for GBR models both in vitro and in vivo. The design of the experiments was clear and well-elaborated. The reviewer was impressed by the proficiency of the work, displaying in vivo functionalities in rabbit GBR, which could lead to larger preclinical animal models in the future.

However, there are some issues the authors could have addressed before accepting for publication.

The reviewer considered the writing as not concise. In the abstract, the novelty of the research should be clearly highlighted. The existing research work covered in the discussion part should be short and concise, as it is where your significance was to be discussed and highlighted. 

Figure 1,4&5 could be combined as one schematic diagram showing the preparation and characterization of the INP.

We appreciate the suggestion of the Rewiever. However, the combined Figure decreases the overall visibility of the results and dramatically modifies the linear comprehension of the text. We suggest the following updated Figure:

Figure 1 -A Preparation and photo-crosslinking of IPN. First, solutions of Si-HPMC and DexMA containing the PIS were mixed together using two syringes connected with a Luer lock. The acidic buffer was then added to the mixture using another syringe connected with a Luer lock. The IPN solution was then ready to be injected and photo-crosslinked (420–480nm). Si-HPMC: silanized hydroxypropyl methylcellulose, DexMA: dextran methacrylate, PIS: photoinitiator solution, IPN: interpenetrating polymer network. B Schematic outline of the preparation of an implant composed of BCP and IPN hydrogel membrane. The cylindrical mold was filled with BCP. Si-HPMC/DexMA solution was injected onto the BCP surface. The solution was photo-crosslinked for 120 s with visible light from a lamp to form the IPN. BCP with IPN and IPN after removal from the BCP were observed with a microscope. Mold with an 8 mm diameter and a 1.5 mm height. Si-HPMC: silanized hydroxypropyl methylcellulose, DexMA: dextran methacrylate, Si-HPMC/DexMA: silanized hydroxypropyl methylcellulose/dextran methacrylate, BCP: biphasic calcium phosphate, IPN: interpenetrating polymer network C : Cytocompatibility assessed by neutral red uptake by L929 cells seeded at 16 000/cm2 after (C1) 24 h in the presence of DexMA polymer extract, (C2) 24 h in the presence of 5 µL/mL of PIS and Si-HPMC, DexMA, or Si-HPMC/DexMA. In order to examine the effect of irradiation, the solutions were exposed to visible light from a lamp for 120 s. The results are presented as means ± SEM (n=3) and analyzed by two-way ANOVA with the Bonferroni post hoc test. Si-HPMC: silanized hydroxypropyl methylcellulose, DexMA: dextran methacrylate, Si-HPMC/DexMA: silanized hydroxypropyl methylcellulose/dextran methacrylate, PIS: photoinitiator solution

What is the position of the editor and the other reviewers ?

The resolution of almost every figure was low, and the font size of texts and legends must be uniform.

The resolution of the figures has been upgraded.

The scale bars must be given in Figure 7&8.

We agree with the reviewer, and scale bars have been added in figure 7 and 8

P values must be calculated among groups in Figure 7&8 C

We agree with the reviewer. We have calculated the P values, and no statistical difference have been evidenced between the groups. Therefore, the P values are not provided on the Figures.

Reviewer 3 Report

Manuscript No.: bioengineering-2025996-peer-review-v1

Title: Injectable in situ interpenetrated polymer network (IPN) hydrogel membrane for guided bone regeneration

Bioengineering

Reviewer’s Decision: Accept after major revision

The authors of this research work describe the injectible hydrogel membrane for guided bone regeneration. The research is significant and should be published in Bioengineering. However, the manuscript must be significantly improved before it can be published. As a result, I recommend accepting the manuscript after significant and satisfactory revisions. The following are the detailed comments:

Critical comments:

  • Structural properties of the injectible hydrogels should be added, if possible.
  • If possible, the morphological behavior of the injectable guided hydrogels should be added.

1.     Title: The title appears confusing; the author has developed an injectible IPN membrane for guided bone tissue engineering, and the authors have studied the biological applications. From the title, it seems that the material was engineered focused on characterizations with bone regeneration application. The title of any research study is to the point so that reader may understand the maximum idea of the research. It is, therefore, recommended to update the title to avoid confusion for the readers.

2.     Abstract: The abstract is a comprehensive summary of the whole research article. The abstract contains numerous grammatical and formatting errors. It is suggested to improve the grammar and English language problems. The abstract section is more introductive and contains methodology information. The methodology and information should be reduced in the informative and methodology section, and add more results outputs with specific biomedical applications. The incomplete information in the abstract may confuse the readers.

3.     Introduction: In the introduction section, material information is detailed, literature on bone tissue or bone regeneration is insufficient, and it is suggested to start with bone regeneration and bone-related system. The introduction and the rest of the manuscript have several grammatical and formatting errors. Please improve the grammar, language, and formatting issues in the manuscript.

a. The page. 02, lines 63-69 “These materials must avoid inflammatory reactions and display good load-bearing capacities against the compressive force of the overlying soft tissue while maintaining sufficient space for tissue regeneration. In addition, clinical handling by physicians during membrane application is also of critical importance. To this end, injectable formulations, which are capable of filling complex defects and rapidly forming a solid membrane upon injection, are particularly appealing, owing to their ease of handling and time- and cost-saving features.” should be written as “These materials shouldn’t cause any inflammatory responses but exhibit strong load bearing capacities against the compressive force of the soft tissue overlying them, and provide adequate space for tissue regeneration. Additionally, the clinical handling done by doctors while applying membranes is crucial. Due to their simplicity of use, quick formation of a solid membrane after injection, and ability to fill complex defects, injectable formulations are particularly appealing in this regard.

b. Page. 02, lines 72-76 “This self-hardening hydrogel has been extensively studied for various applications such as cartilage repair, bone regeneration or drug delivery for intervertebral discs. In addition, we have demonstrated the ability of this cross-linked polymer to act as a physical barrier against cell invasion in a periodontal defect in dogs.” should be written as “Numerous applications of this self-hardening hydrogel, including drug delivery for intervertebral discs, bone regeneration, and cartilage repair, have been thoroughly investigated. Additionally, in a canine periodontal defect, we have shown that this cross-linked polymer can function as a physical barrier against cell invasion.

4.     References: The manuscript lacks the literature citation of some highly interesting, most recent relevant works; thus, the references are not up to date. Too many references have been given for a research article that may question the novelty of research work that may give an impression that so many people have reported the research already. These citations will help to explain polysaccharide-based hydrogels and scaffolds for bone regeneration. In this regard, the author should refer to some of the most recent papers on hydrogel, such as

·       Mannan, H. et al. (2022). Sodium alginate-f-GO composite hydrogels for tissue regeneration and antitumor applications. International Journal of Biological Macromolecules208, 475-485.

·       Ashammakhi, N., et al. (2022). Multifunctional Bioactive Scaffolds from ARX-g-(Zn@ rGO)-HAp for Bone Tissue Engineering: In Vitro Antibacterial, Antitumor, and Biocompatibility Evaluations. ACS Applied Bio Materials.

·       Tang, B. et al. (2021). Strontium Laminarin polysaccharide modulates osteogenesis-angiogenesis for bone regeneration. International journal of biological macromolecules181, 452-461.

·       Stojanović, G. M. et al. (2022). Bioactive scaffold (sodium alginate)-g-(nHAp@ SiO2@ GO) for bone tissue engineering. International Journal of Biological Macromolecules222, 462-472.

5.     Materials and methods: This section is missing; please add it; otherwise, it may confuse the readers.

“Well described”

6.     Results and Discussions: The following issue must be taken into consideration.

a.     It is recommended to add cell morphology and adherence, if possible.

b.     The figure caption and style should be identical throughout the manuscript to keep the continuity of the work.

c.     Statistical significance is required in Fig. 3, Fig. 5, Fig. 7C, and Fig 8C.

7.     Conclusions: The conclusion section is the most important summary of a research article, and it should be based on the conclusion for the conclusion. The conclusion should be based on comparing the different used formulations by comparing the best result output. However, it is recommended to revise the conclusion section based on conclusions as the conclusion section contains more information and methodology rather than comparisons of the different formulations.

8.     As per the comments given for the results and description.

In summary, the reported work has significant value; however, a major and thorough improvement/correction of language, grammar, syntax, etc., is necessary to improve the paper’s quality and make it publishable in Bioengineering.

Author Response

R3

Manuscript No.: bioengineering-2025996-peer-review-v1

Title: Injectable in situ interpenetrated polymer network (IPN) hydrogel membrane for guided bone regeneration

Bioengineering

Reviewer’s Decision: Accept after major revision

The authors of this research work describe the injectible hydrogel membrane for guided bone regeneration. The research is significant and should be published in Bioengineering. However, the manuscript must be significantly improved before it can be published. As a result, I recommend accepting the manuscript after significant and satisfactory revisions. The following are the detailed comments:

Critical comments:

  • Structural properties of the injectible hydrogels should be added, if possible.
  • If possible, the morphological behavior of the injectable guided hydrogels should be added.

We agree with the reviewer, but gel morphological behavior has not been evaluated.

  1. Title:The title appears confusing; the author has developed an injectible IPN membrane for guided bone tissue engineering, and the authors have studied the biological applications. From the title, it seems that the material was engineered focused on characterizations with bone regeneration application. The title of any research study is to the point so that reader may understand the maximum idea of the research. It is, therefore, recommended to update the title to avoid confusion for the readers.

We didn’t fully understand the reviewer’s comment. We have slightly modified the title.

  1. Abstract: The abstract is a comprehensive summary of the whole research article. The abstract contains numerous grammatical and formatting errors. It is suggested to improve the grammar and English language problems. The abstract section is more introductive and contains methodology information. The methodology and information should be reduced in the informative and methodology section, and add more results outputs with specific biomedical applications. The incomplete information in the abstract may confuse the readers.

We agree with the reviewer, and only 2 lines of methods have been kept according to the journal guidelines.

  1. Introduction:In the introduction section, material information is detailed, literature on bone tissue or bone regeneration is insufficient, and it is suggested to start with bone regeneration and bone-related system. The introduction and the rest of the manuscript have several grammatical and formatting errors. Please improve the grammar, language, and formatting issues in the manuscript.
  2. The page. 02, lines 63-69 “These materials must avoid inflammatory reactions and display good load-bearing capacities against the compressive force of the overlying soft tissue while maintaining sufficient space for tissue regeneration. In addition, clinical handling by physicians during membrane application is also of critical importance. To this end, injectable formulations, which are capable of filling complex defects and rapidly forming a solid membrane upon injection, are particularly appealing, owing to their ease of handling and time- and cost-saving features.” should be written as “These materials shouldn’t cause any inflammatory responses but exhibit strong load bearing capacities against the compressive force of the soft tissue overlying them, and provide adequate space for tissue regeneration. Additionally, the clinical handling done by doctors while applying membranes is crucial. Due to their simplicity of use, quick formation of a solid membrane after injection, and ability to fill complex defects, injectable formulations are particularly appealing in this regard.”

We agree with the reviewer, and modifications have been made accordingly

  1. Page. 02, lines 72-76 “This self-hardening hydrogel has been extensively studied for various applications such as cartilage repair, bone regeneration or drug delivery for intervertebral discs. In addition, we have demonstrated the ability of this cross-linked polymer to act as a physical barrier against cell invasion in a periodontal defect in dogs.” should be written as “Numerous applications of this self-hardening hydrogel, including drug delivery for intervertebral discs, bone regeneration, and cartilage repair, have been thoroughly investigated. Additionally, in a canine periodontal defect, we have shown that this cross-linked polymer can function as a physical barrier against cell invasion.”

We agree with the reviewer, and modifications have been made accordingly.

  1. References:The manuscript lacks the literature citation of some highly interesting, most recent relevant works; thus, the references are not up to date. Too many references have been given for a research article that may question the novelty of research work that may give an impression that so many people have reported the research already. These citations will help to explain polysaccharide-based hydrogels and scaffolds for bone regeneration. In this regard, the author should refer to some of the most recent papers on hydrogel, such as
  • Mannan, H. et al. (2022). Sodium alginate-f-GO composite hydrogels for tissue regeneration and antitumor applications. International Journal of Biological Macromolecules208, 475-485.
  • Ashammakhi, N., et al. (2022). Multifunctional Bioactive Scaffolds from ARX-g-(Zn@ rGO)-HAp for Bone Tissue Engineering: In Vitro Antibacterial, Antitumor, and Biocompatibility Evaluations. ACS Applied Bio Materials.
  • Tang, B. et al. (2021). Strontium Laminarin polysaccharide modulates osteogenesis-angiogenesis for bone regeneration. International journal of biological macromolecules181, 452-461.
  • Stojanović, G. M. et al. (2022). Bioactive scaffold (sodium alginate)-g-(nHAp@ SiO2@ GO) for bone tissue engineering. International Journal of Biological Macromolecules222, 462-472.

References have been added accordingly to the reviewer's suggestion (e.g., ref 6 and 8)

  1. Materials and methods: This section is missing; please add it; otherwise, it may confuse the readers.

We agree with the reviewer, and modifications have been made accordingly

“Well described”

  1. Results and Discussions:The following issue must be taken into consideration.
  2. It is recommended to add cell morphology and adherence, if possible.

We agree with the reviewer, but cell morphology has not been evaluated yet. It is an important parameter and will be checked in further experiments.

  1. The figure caption and style should be identical throughout the manuscript to keep the continuity of the work.

The figures have been upgraded.

  1. Statistical significance is required in Fig. 3, Fig. 5, Fig. 7C, and Fig 8C.

We agree with the reviewer, but no statistical differences have been evidenced.

  1. Conclusions:The conclusion section is the most important summary of a research article, and it should be based on the conclusion for the conclusion. The conclusion should be based on comparing the different used formulations by comparing the best result output. However, it is recommended to revise the conclusion section based on conclusions as the conclusion section contains more information and methodology rather than comparisons of the different formulations.
  2. As per the comments given for the results and description.

In summary, the reported work has significant value; however, a major and thorough improvement/correction of language, grammar, syntax, etc., is necessary to improve the paper’s quality and make it publishable in Bioengineering.

A professional native English corrector has been subcontracted to correct and improve the quality of the manuscript.

Round 2

Reviewer 3 Report

All the comments have been addressed successfully and it can be accepeted in present form.

Author Response

Dear reviewer

Thank you for your decision.

We did all the corrections.

Regards